# OpenReview forum: "RM-Bench: Benchmarking Reward Models of Language Models with Subtlety and Style"
_ICLR.cc/2025/Conference — ICLR 2025 Oral_

### Official Review · Reviewer_9kQK · 2024-10-19

**Soundness:** 3
**Presentation:** 4
**Contribution:** 3
**Rating:** 8
**Confidence:** 4

**Summary:**

This paper introduces a benchmark for reward models that is designed to be robust to subtleties in generated text. Previous reward model benchmarks have used weaker models to generate rejected responses, and stronger models for the chosen responses. Such a benchmark could be exploited by a reward model that prefers properties of responses unrelated to what the benchmark is intended to evaluate, such as the style of text.

Instead of using stronger and weaker models, RM-Bench uses one model to generate the chosen and rejected responses, where the rejected responses are flawed in some way (e.g., they were generated by a model that had previously been jailbroken). Style is also controlled via prompting, and the benchmark covers three styles (concise, detailed and markdown). The accuracy of RM-Bench is evidenced by showing correlation between reward models that score higher on RM-Bench and improved performance of policy models on evaluations such as GSM8k and HumanEval+.

The main takeaway is that even the best publicly available reward models have imperfect performance on RM-Bench, so there is plausibly room for significant improvements in reward modeling. The paper also shows that DPO models outperform some sequence classifiers on RM-Bench, and that the importance of style has likely gone underappreciated in reward model evaluation.

The Appendix gives evidence that RM-Bench is more accurate than prior work by showing the weaker correlation for benchmark performance of the policy model and performance on Reward Bench.

**Strengths:**

**Originality**: I'm not aware of prior work that benchmarks reward models while controlling for style. Similar ideas have been discussed generally in RL, but not used to design better reward model benchmarks. To the best of my knowledge the contributions are original.

**Quality**: The method is sufficiently rigorous for the results to be reproduced. The way the paper evaluates the accuracy of RM-Bench is sound, and the three main takeaways from the paper are well evidenced. For example, Appendix K shows the authors covered many popular publicly available reward models, which supports the claim that progress in reward modeling is necessary.

**Clarity**: The paper is clearly written and the figures are not confusing. The authors introduce the problem, background and their contributions well.

**Significance**: I find the results significant. They point to what seems to be an important issue in a previous evaluation and improve upon it by mitigating that issue. They then show the improved benchmark is correlated with downstream policy model performance. The other takeaways of the paper are also significant in their own right, such as benchmarking the reward modeling of models trained with DPO, but these are of course less significant than the main contribution of the paper.

**Weaknesses:**

- As far as I could tell the exact data used in the paper is not available. It would be good to have a 'canonical' version of RM-Bench on HuggingFace or somewhere similar to make reproducing the results easier.
- The paper does not show that the reason RM-Bench is more accurate is because you control for subtleties in the text like style. Although this seems likely, showing evidence that this isn't because of other differences between RM-Bench and Reward Bench (e.g. by ablating the control for style bias) would increase my soundness score to a 4.
- It would be interesting to see the correlations separately for each of the benchmark types (math, code, safety).

**Questions:**

Do you plan to have a version of the data you used in your experiments up on HF for the camera-ready version of the paper?

---

> ### Author Response · Authors · 2024-11-22
> **Response to Reviewer 9kQk**
>
> We are very glad that you recognize the originality of our benchmark as the first to evaluate reward models while controlling for style. We are grateful for your positive feedback on the significance of our results. We appreciate your detailed feedback and suggestions for improvement. We treasure the opportunity to address your concerns and improve our work.
>
> ---
>
> > **Weaknesses 1 and Question 1:** The open-source plan of RM-Bench.
>
> We definitely plan to release the data used in our experiments on HuggingFace along with the corresponding evaluation code on GitHub.
>
> As devoted supporters of the open-source community, we will include not only the data but also the model weights files used in Sections 4 and 5 to facilitate reproducibility and further research.
>
> Due to the anonymity policy, we cannot include the link to the data in the submission version. However, we will ensure that the link to the data is included in the camera-ready version of the paper upon acceptance.
>
> ---
>
> > **Weakness 2:** Ablation study to show why RM-Bench is more accurate.
>
> We appreciate the suggestion. We have included an ablation study in Appendix O of the revised version of our paper.
>
> In short, we evaluated the correlations with policy model performance based on Easy, Normal, and Hard Accuracy, respectively:
>
> - **Easy Accuracy**: Represents the scenario without substance or style control, with a correlation of **0.07**.
> - **Normal Accuracy**: Represents the scenario with substance control applied, with a correlation of **0.30**.
> - **Hard Accuracy**: Represents the scenario with both substance and style control applied, with a correlation of **0.45**.
>
>
> These results show a consistent increase in correlation with the application of substance and style control. This testifies that the reason RM-Bench is more accurate is because of the substance control and style control design.
>
> We will ensure this information is included in the camera-ready version of the paper if it is accepted.
>
> ---
>
> > **Weakness 3:** Correlation results for each of the benchmark types (math, code, safety).
>
> We have included the correlation results for each benchmark type (math, code, safety) in Appendix P. The results demonstrate consistent correlation across all benchmark types.
>
> We will ensure this information is included in the camera-ready version of the paper if it is accepted

---

> > ### Comment · Reviewer_9kQK · 2024-11-25
> >
> > Thank you for the response. I appreciate the inclusion of Appendix O and P, and believe it addresses the weaknesses I mentioned. As the rating I gave is already an 8, I don't feel the need to revise my score, but I do believe these improvements make the paper better.

---

> > > ### Author Response · Authors · 2024-11-26
> > >
> > > Thank you for your active feedback and engagement during the rebuttal process. We appreciate your endorsement!

---

### Official Review · Reviewer_RBmr · 2024-10-31

**Soundness:** 3
**Presentation:** 3
**Contribution:** 3
**Rating:** 8
**Confidence:** 3

**Summary:**

This paper introduces a new benchmark, called RM-Bench, to evaluate reward models. The authors state that an ideal reward model should be able to identify subtle differences in content quality (correctness) and consistently reward better responses, regardless of stylistic variations. They argue that existing benchmarks, which rely on comparing responses generated by LLMs of differing power, fail to fully capture these aspects of reward model evaluation.

RM-Bench addresses this by constructing a dataset where both chosen and rejected responses are generated by the same LLM, gpt-4o (except for safety, where responses are generated using different models to ensure safety violations). They use (i) domain-specific techniques to introduce subtle errors to the rejected responses, and (ii) prompting the LLM to generate response variants with differing levels of detail and markdown formatting. Their main contribution is the introduction of the style-substance evaluation matrix and the corresponding {easy, normal, hard} accuracy metrics that allow a more granular analysis of how style influences reward model predictions.

By evaluating a wide range of reward models on RM-Bench, they demonstrate that a wide range of reward models have a style bias (a high easy accuracy but a low hard accuracy). They also empirically show that the performance of a reward model on RM-Bench (moderately) correlates with the performance of the resulting aligned policy model (under style-controlled evaluation and across downstream tasks).

**Strengths:**

The use of style-controlled generation of responses for building the evaluation dataset is a simple but structured and original idea to directly assess the style bias in reward models. It is generic in the sense that it can also be applied to existing datasets with verbose responses (or the model can be asked to alter the response such that they are more verbose and with formatting). Style-substance evaluation matrix enables to quantify the degree of the bias and its impact on reward predictions.

The experiments are comprehensive and demonstrate that it is a challenging benchmark for the current reward models, exposing their weaknesses with respect to style bias and robustness. This is a significant result (but may be of limited scope; see below). The performances of the aligned policy models, measured by their style-control scores and how well they perform in downstream tasks, correlate positively with the performance of the corresponding reward models on RM-Bench. This correlation, while not strong, is nonetheless better than that observed with RewardBench, which does not explicitly focus on style bias and relies on preference datasets with responses that are human curated or generated using a strong and a weak LLM pair. However, there may be multiple parameters in play here and a clarification may be needed.

**Weaknesses:**

In the paper, the authors construct the RM-Bench dataset in two steps: (i) generate domain specific preference pairs as described in sections 3.1-3.3, and (ii) add style-controlled variants of these pairs as described in section 3.4. Adding style-controlled variants (step (ii)) is essential to understand the style bias and the main focus of the paper, but its contribution to (average) accuracy and the correlation with the performance of the aligned policy models is unclear. In Table 3, we can see that the normal accuracy is close to the average accuracy, which may hint that accuracy based on (y_c^{L,M}, y_r^{L, M}) pairs alone could be be a good indicator. To better understand the role of the style-controlled generation, the authors should provide these numbers and compare the results in sections 4.2 (DPO) and 5 (correlation with the policy model) with RM-Bench performance based on y^{L, M} only.

In section 4.3, the correctness and verbosity scores of the examples (from Nemotron-4-340B-Reward model) are conditioned on the prompt. The scores of responses with different prompts may not be comparable. This makes it difficult to interpret the presented results. What is the behavior of the model when you use the accuracy r(y_c) > r(y_r) is used for both correctness and verbosity scores instead?

**Questions:**

- In section 3.1, after human filtering the number of examples with subtle errors drop significantly (519->183). Is this mainly due to chosen responses (generated by gpt-4o) not being correct or many-shot jailbreak technique failing to produce responses with injected factual errors? Can you elaborate more on the effectiveness of the approach?

- (section 3.2) HumanEvalPack contains both the correct and buggy (human written) solutions. Does using multiple-sampling to generate rejected samples bring an advantage over using the existing ones when assessing the performance of the reward models for the corresponding domain?

- (section 3.4) What are the prompts used to generate the plain text and concise prompts?

- (section 4.2) How does the DPO model perform when you use the hard instead of average accuracy? The DPO paper initializes the reference policy by maximizing the likelihood of preferred completions when the reference model is not available. Setting \pi_{ref}(y|x) = 1 assumes a uniform distribution over all possible responses for the reference model. You may want to compare with the former as well.

- Section I: The instruction asks the model to introduce one error, but there are examples with several errors. Could this be causing over filtering?

- The natural approach to improve the performance of a reward model in RM-Bench would be to augment the training (preference) dataset of the reward model using style-controlled generation as explained in section 3.4 (with proper prompts). We expect the model to have a higher (hard) accuracy and a higher score in this case. However, this may not lead to an improvement in the performance of the aligned policy model that uses such a reward model (hacking the benchmark instead). In section 5.1, you show a moderate positive correlation, but the reward models there do not explicitly take style bias into consideration. Could you please elaborate on this?

---

> ### Author Response · Authors · 2024-11-22
> **Response to Reviewer RBmr (Part 1/2)**
>
> We appreciate that the reviewer recognizes the originality and significance of our work. We thank the reviewer for their detailed feedback and constructive suggestions. Here, we treasure the opportunity to address your concerns and improve the quality of our work.
>
> Before addressing the specific points raised by the reviewer, we would like to point out that the contribution of RM-Bench is not limited to style-control (robustness) but also to substance-control (sensitivity to substance change, like the comparison between $y_c^{L,M}$ and $y_r^{L,M}$) and a more fine-grained evaluation method (style-substance evaluation matrix) compared to existing benchmarks like RewardBench. Thus, the insights we found in Table 4, Section 5 are not solely based on style-control but also on substance-control.
>
> ---
>
> > **Weaknesses 1:** In Table 4, Section 5, the RM-Bench performance should be compared based on $y^{L, M}$ only.
>
> We further provide the results of RM-Bench performance based on $y^{L, M}$ only in Table 4, Section 5 in Appendix N in the revised submission.
>
> In short, for Table 4 (DPO vs. Sequential Classifier), the conclusion still holds. The DPO model outperforms the Sequential Classifier in RM-Bench. When the reference model is not available, the DPO model performance is degraded, indicating the importance of the reference model in DPO.
>
> For Section 5, the correlation with the policy model performance based on $y^{L, M}$ only is also provided in Appendix N.2. The correlation is weaker than the correlation based on hard accuracy, which indicates the importance of style-control in the evaluation of reward models.
>
> Additionally, we would like to kindly refer you to Appendix O, where we conduct a comprehensive ablation study to investigate the impact of style-control and substance-control on the correlation with the policy model. The results show that both style-control and substance-control are important to provide a better correlation with the policy model performance.
>
> ---
>
> > **Weaknesses 2:** What is the behavior of the model when you use the accuracy $r(y_c) > r(y_r)$?
>
> Here we provide the results of Reward Model Accuracy by comparing $y_c^\varnothing$ and $y_r^\text{L}$ solely based on the correctness score in the following table. When the correctness score of $y_c^\varnothing$ is higher than $y_r^\text{L}$, the data point is considered correct, as $y_c^\varnothing$ is the correct response while $y_r^\text{L}$ is the response with errors.
>
> |                | Chat  | Safety | Math  | Code  |
> |----------------|-------|--------|-------|-------|
> | Acc. (correctness score) | 0.415 | 0.726  | 0.560 | 0.557 |
>
> These results support the same conclusion we made in Section 4.3: the correctness score provided by the Nemotron-4-340B-Reward model is still far from satisfactory, except for the Safety domain. This indicates that the model still fails to separate correctness from other factors like verbosity when assigning correctness scores.

---

> > ### Author Response · Authors · 2024-11-22
> > **Response to Reviewer RBmr (Part 2/2)**
> >
> > > **Question 1:** The number of examples with subtle errors drops significantly (519 → 183).
> >
> > The main reason for this drop is the limited success of the many-shot jailbreak technique in producing responses with injected factual errors. Here are the details of the filtering process:
> >
> > - Reason 1: The error injection process is not successful -> filter out 183 samples
> > - Reason 2: The correct information still remains in the rejected responses -> filter out 66 samples
> > - Reason 3: The format of the response is not desired -> filter out 31 samples
> >
> > ---
> >
> > > **Question 2:** Compare multiple-sampling generated rejected samples with existing human-written buggy solutions.
> >
> > We would like to clarify that there are two main advantages of using multiple-sampling to generate rejected samples:
> >
> > 1. **Scalability**: The multiple-sampling technique allows us to easily expand the dataset with more rejected samples and scale to other datasets that provide test cases. This minimizes human effort.
> > 2. **Alignment to Practical Use**: The buggy code generated by the multiple-sampling technique is produced by the policy model, which reflects the model's current shortcomings. However, human-written buggy solutions may reflect human shortcomings that do not align with the model's shortcomings. Considering that the reward model is used to help the policy overcome the model's shortcomings, the multiple-sampling technique is more aligned with the practical use of the reward model.
> >
> > ---
> >
> > > **Question 3:** What are the prompts used to generate the plain text and concise prompts?
> >
> > Please refer to the general response.
> >
> > ---
> >
> > > **Question 4:** How does the DPO model perform when you use the hard instead of average accuracy?
> >
> > Here is Table 4 based on hard accuracy:
> >
> > | Model                         | HH-RLHF | StackExchange | Nectar | Chatbot Arena 2023 |
> > |-------------------------------|---------|---------------|--------|--------------------|
> > | DPO (Ref. Model Free)         | 16.2    | 15.2          | 13.7   | 14.7               |
> > | Sequence Classifier           | 49.7    | 23.7          | 45.4   | 15.5               |
> > | DPO (With Ref. Model)         | 69.6    | 37.2          | 25.4   | 29.5               |
> >
> > The observation is consistent with the conclusion in Table 4 of Section 4.2.
> >
> > In addition, we would like to clarify that there would be many interesting comparisons between DPO and Sequential Classifiers with the substance-style evaluation matrix in each domain. However, considering that our paper is not focused on providing a comprehensive comparison between DPO and Sequential Classifiers, we leave this as future work. (We will release the detailed substance-style evaluation matrix for these 12 reward models to facilitate future research.)
> >
> > ---
> >
> > > **Question 5:** The instruction asks the model to introduce one error, but there are examples with several errors. Could this be causing over-filtering?
> >
> > No, the error number is not the reason. Please refer to the response to Question 1.
> >
> > ---
> >
> > > **Question 6:** In Section 5.1, the tested reward models do not explicitly take style bias into consideration; the correlation is moderate.
> >
> > First, we would like to clarify that although the training data here does not explicitly take style bias into consideration, style bias inherently exists in the training data.
> >
> > This can be verified by the token number ratio between the chosen response and the rejected response:
> > - For **Chatbot Arena 2023**, the token number ratio between chosen and rejected responses is $3.18$.
> > - For **HH-RLHF**, the token number ratio between chosen and rejected responses is $1.37$.
> >
> > This difference shows that style bias (especially length bias) is much more significant in *Chatbot Arena 2023* than in *Anthropic HH/RLHF*.
> >
> > This difference reflects on the performance of the reward model in RM-Bench. As Figure 4 shows, the reward model trained with *Anthropic HH/RLHF* has a better Chat-Hard performance than the reward model trained with *Chatbot Arena 2023*.
> >
> > It also reflects on the policy model's performance in the Auto Arena Hard task. As Figure 4 shows, the policy model trained with *Anthropic HH/RLHF* has better style-control performance than the policy model trained with *Chatbot Arena 2023*.
> >
> > Second, we would like to point out that the correlation in Section 5.1 is not moderate but strong. This is because, as the performance improves in Chat-Hard, the policy model is consistently better in the Auto Arena Hard task. If we calculate the rank Spearman correlation, it is 1.0 with a p-value of 0.0.

---

> > > ### Comment · Reviewer_RBmr · 2024-11-25
> > >
> > > Thank you for the answers. Please see below for additional comments.
> > >
> > > Q1: Should you then highlight in the paper the shortcoming of the error injection approach? This limits the scalability that you mention in answer to Q2 (and also in the paper).
> > >
> > > Q2: I agree with these advantages, the question was however about the comparison of the negative examples that are known to be good (human generated) and sampled from the model and its impact on the results that you provide. Given that "the correctness of the responses is verified by human annotators to ensure high-quality data across all domains" for the chat domain and the number of samples are comparable (Table 2), it would be good to justify the approach empirically.
> > >
> > > Q3: The results on the Nectar dataset seem to contradict the general conclusion in Section 4.2 (in particular, given that the focus is on substance rather than style). It would be good to revise it accordingly.

---

> > > > ### Author Response · Authors · 2024-11-26
> > > > **Response (Part 1/2)**
> > > >
> > > > Thank you for your feedback. We appreciate the opportunity to address your concerns and improve the quality of our work. Here, we provide additional analysis and clarification to address your comments.
> > > >
> > > > ---
> > > >
> > > > > **Weakness 2: Relationship Between Verbosity and Correctness Scores Using the Binary Correctness Measure**
> > > >
> > > > We appreciate the reviewer's feedback. To demonstrate the effect of verbosity on the correctness score, we present the results of the accuracy based on comparisons between $y_c^\varnothing$ and $y_r^\varnothing$ and between $y_c^\varnothing$ and $y_r^\text{L}$. These comparisons are shown in the table below. The difference in accuracy illustrates how changing the rejected response from concise ($y_r^\varnothing$) to verbose ($y_r^\text{L}$) affects the correctness score assigned by the reward model.
> > > >
> > > > |                   | $y_c^\varnothing$ vs. $y_r^\varnothing$ | $y_c^\varnothing$ vs. $y_r^\text{L}$ | Accuracy Decrease (↓) |
> > > > |-------------------|---------------------------|---------------------------|-----------------------|
> > > > | Chat              | 0.798                     | 0.415                     | ↓0.383                |
> > > > | Safety            | 0.939                     | 0.726                     | ↓0.213                |
> > > > | Math              | 0.595                     | 0.560                     | ↓0.035                |
> > > > | Code              | 0.605                     | 0.557                     | ↓0.048                |
> > > >
> > > > The table demonstrates a significant decrease in binary accuracy when the rejected response changes from concise ($y_r^\varnothing$) to verbose ($y_r^\text{L}$). This suggests that higher verbosity is more likely to increase the correctness score of the rejected response, leading to an even higher correctness score than the chosen response.
> > > >
> > > > We hope this additional analysis clarifies the impact of verbosity on the correctness score and their relationship. If you have further questions or require additional information, please let us know.

---

> > > > > ### Author Response · Authors · 2024-11-26
> > > > > **Response (Part2/2)**
> > > > >
> > > > > > **Question 1: Highlighting the Shortcomings of the Error Injection Approach**
> > > > >
> > > > > We appreciate the reviewer's feedback. After revision, we have included a discussion in Section 3.1 and Appendix D to highlight the shortcomings of the error injection approach.
> > > > > This shortfall stems from the limited in-context examples used in jailbreaks due to computational and cost constraints.
> > > > > In the future, we will explore to scale the number of in-context examples used in jailbreaks to hundreds   to alleviate this limitation when we get sufficient computational resources.[1]
> > > > >
> > > > > [1] Many-shot jailbreaking https://www.anthropic.com/research/many-shot-jailbreaking
> > > > >
> > > > >
> > > > >
> > > > > ---
> > > > >
> > > > > > **Question 2: Comparison of Human-Generated and Model-Sampled Examples**
> > > > >
> > > > > We appreciate the reviewer’s feedback. Below, we further provide an empirical comparison between the human-written samples from HumanEvalPack and those generated using the multiple-sampling technique.
> > > > >
> > > > > Before presenting the comparison, we would like to clarify the following points:
> > > > > - Directly using human-written correct/buggy solutions from HumanEvalPack to form chosen/rejected pairs **is identical to the approach used in RewardBench** for their code split.
> > > > > - Solutions in HumanEvalPack are extremely concise and consist solely of code snippets, with no explanations or comments. Additionally, no style variations of solutions are provided, unlike in RM-Bench.
> > > > >
> > > > > Given this, we treat the correct human-written solutions from HumanEvalPack as concise correct responses ($y_c^\varnothing$) and the buggy solutions as concise rejected responses ($y_r^\varnothing$). We evaluate the top 10 reward models on $y_c^\varnothing$ and $y_r^\varnothing$ from HumanEvalPack and compare these accuracies with those obtained from RM-Bench. The results are presented below:
> > > > >
> > > > > | Model Name                                      | $y_c^\varnothing$ vs. $y_r^\varnothing$ in RM-Bench | $y_c^\varnothing$ vs. $y_r^\varnothing$ in HumanEvalPack | Data Leakage |
> > > > > |-------------------------------------------------|---------|---------|----------------|
> > > > > | openbmb/Eurus-RM-7b                             | 0.5382  | 0.9167  | [not confirmed]  |
> > > > > | NCSOFT/Llama-3-OffsetBias-RM-8B                 | 0.5014  | 0.9254  | [not confirmed]  |
> > > > > | sfairXC/FsfairX-LLaMA3-RM-v0.1                  | 0.5042  | 0.9342  | [not confirmed]  |
> > > > > | LxzGordon/URM-LLaMa-3-8B                        | 0.4844  | 0.8860  | [confirmed]        |
> > > > > | LxzGordon/URM-LLaMa-3.1-8B                      | 0.5127  | 0.8816  | [confirmed]        |
> > > > > | Skywork/Skywork-Reward-Llama-3.1-8B             | 0.5326  | 0.9737  | [confirmed]        |
> > > > > | internlm/internlm2-20b-reward                   | 0.5014  | 0.9605  | [not confirmed]  |
> > > > > | internlm/internlm2-7b-reward                    | 0.5042  | 0.9254  | [not confirmed]  |
> > > > > | Ray2333/GRM-llama3-8B-sftreg                    | 0.5156  | 0.9430  | [confirmed]        |
> > > > > | Ray2333/GRM-llama3-8B-distill                   | 0.5042  | 0.9386  | [confirmed]        |
> > > > > | Ray2333/GRM-Llama3-8B-rewardmodel-ft            | 0.5156  | 0.9430  | [confirmed]        |
> > > > >
> > > > > As shown in the table, the accuracy in HumanEvalPack is significantly higher than the accuracy in RM-Bench.
> > > > > This observation could be due to the knowing data contamination issue in RewardBench, as the authors of RewardBench stated in the leaderboard **"⚠️ Many of the top models were trained on unintentionally contaminated, AI-generated data."**.[1,2]
> > > > > This possible data leakage potentially lead to a higher accuracy in HumanEvalPack than in RM-Bench.
> > > > > It also highlights the significance of RM-Bench in providing a clean and reliable evaluation dataset by generating new responses with style variations from the latest powerful LLMs.
> > > > >
> > > > >
> > > > > [1] RewardBench leaderboard https://huggingface.co/spaces/allenai/reward-bench
> > > > >
> > > > > [2] natolambert/skyworks-rewardbench-contamination.md https://gist.github.com/natolambert/1aed306000c13e0e8c5bc17c1a5dd300
> > > > >
> > > > > ---
> > > > >
> > > > > > **Question 3: Outlier in the Hard Accuracy Setting with Nectar Dataset**
> > > > >
> > > > > We appreciate the reviewer’s feedback. The results of the outlier caused by the Nectar dataset under the Hard Accuracy setting have been included in Appendix Q of the revised submission. Additionally, we have updated the content in Section 4.2 to reflect this outlier in the Nectar dataset.
> > > > >
> > > > > As our paper does not focus on providing a comprehensive and in-depth comparison between DPO and Sequential Classifiers, we propose this as a future research direction. To support further research, we will release the detailed substance-style evaluation matrix for the 12 reward models, as mentioned in our previous response.
> > > > >
> > > > > ---
> > > > >
> > > > > If you have any further questions or need additional information, please let us know. We are more than happy to assist you. Thank you for your valuable feedback and suggestions.

---

> > > > > > ### Comment · Reviewer_RBmr · 2024-11-28
> > > > > >
> > > > > > Thank you for the responses and addressing the concerns, I updated my rating accordingly.

---

> > > > > > > ### Author Response · Authors · 2024-11-29
> > > > > > >
> > > > > > > Thank you for your thoughtful feedback and active engagement throughout the rebuttal process. We sincerely appreciate your support and endorsement!

---

> > ### Comment · Reviewer_RBmr · 2024-11-25
> >
> > Thank you for the additional experiments. For (w2), the comment was about seeing the relationship between the verbosity and correctness scores when you use the binary correctness measure (which is comparable between examples with different prompts). Your response doesn't seem to answer this.

---

### Official Review · Reviewer_SzZ1 · 2024-11-03

**Soundness:** 3
**Presentation:** 3
**Contribution:** 3
**Rating:** 8
**Confidence:** 4

**Summary:**

In this submission, the authors have introduced RM-BENCH, a benchmark for evaluating reward models that: 1. evaluates reward models' sensitivity to subtle content differences and 2. tests resistance to style biases like length and markdown formatting.

**Strengths:**

- The paper is well-motivated and offers various prompts to explicitly control for style bias, in contrast to the RewardBench benchmark.
- It also establishes a clear methodology for evaluating biases, which could be extended to other categories or used to add tasks focused on reasoning and safety. In short, the overall methodology appears scalable, though it may require some human-level judgment.

**Weaknesses:**

I feel that the paper in its current state has a lot of scope for improvement, and hence I am leaning towards a borderline reject. I might be willing to increase my score after the rebuttal period with the authors if they are able to improve upon the following areas:

- Currently, the benchmark relies heavily on gpt4o. I am aware that the authors discussed this in the Appendix's Limitations section, but I strongly feel that, to make the benchmark ready for deployment, it should be diverse and include responses from more SOTA models like Claude, Gemini, etc.
- The results shown in Table 4 and the correlation analysis in Section 5 and the Appendix rely on the Tulu-v2.5 model. I would like to see the same analysis on at least one more model (maybe from Qwen or LLaMA family) before drawing conclusions on correlation analysis.
- While the authors explain that they use gpt4o to create different versions of the same content (short vs. long, with and without special formatting), they do not provide enough details about how exactly they do this or how they ensure that the different versions maintain the same meaning. These missing details about their exact process—such as the specific prompt given to gpt4o for summarizing responses or removing markdown, or how they check the quality of generated responses (after removing markdown)—should be included in the Appendix.
- I would also like to see more details on length control in Table 2. For example, what are the target length ranges for each category when selecting normal vs. long responses, and how do you ensure that, for concise responses (both correct and rejected), the model retains the core information (fact or code snippet) even after summarization?

**Questions:**

1. How much overlap is there between the samples from RM-Bench and RewardBench?
2. On Page 8, Lines 426–431, the authors state that “the reward model fails to detect subtle differences between chosen and rejected responses, which causes significant overlap in the scatterplot.” However, as seen in Table 2, samples from Math and Code tasks have nearly identical average tokens for short and long responses (for both correct and rejected ones). So, wouldn’t one expect significant overlap in the plot? I think the authors should clarify the minimum, maximum, and median lengths of samples across all tasks and style types.
3. In Figure 4, what do the four data points signify? Do they represent models trained on four different preference datasets? If so, the authors should clarify which data point corresponds to which preference dataset.
4. Related to point 3 above: Figure 4 assesses Chat Hard accuracy. If the preference data is StackExchange, then I would also like to see a similar graph with the x-axis as Code Hard or Code Normal accuracy. Additionally, I would like to see a similar graph on one more base language model besides Tulu to verify the claims made.
5. In Figure 8 of Appendix H, does the y-axis represent score or relative accuracy?
6. Typos:
    1. Page 4 Line 180: “LLM we used here is” → “The LLM that we used here is”
    2. Page 4 Lines 182-183: A sentences seems to be partially omitted. Please clarify what the authors are trying to convey there.
    3. Page 10 Line 504: “It provides” → “They provide”
    4. Page 10 Line 503: “generally used to” → “generally used ~~to~~”
    5. Page 15 Line 777: “Border Impacts” → “Broader Impact”
    6. Page 15 Line 786: “build” → “built”
    7. Page 16 Line 815: “open-end” → “open-ended”

---

> ### Author Response · Authors · 2024-11-22
> **Response to Reviewer SzZ1 (Part 1/2)**
>
> Thank you for the detailed and constructive feedback! We treasure the opportunity to address your concerns and improve our work.
>
> ---
>
> > **Weakness 1:** RM-Bench should include responses from more SOTA models like Claude, Gemini, etc.
>
> Following your suggestion, we developed a new version of RM-Bench using the Gemini-1.5-Pro model. This version follows the exact same data curation pipeline outlined in our paper. We have reevaluated the top-10 models in Table 3 using the Gemini-based RM-Bench. Here are the results:
>
> | Model Name                                    | Average Acc (%) |
> |-|-|
> | Skywork/Skywork-Reward-Llama-3.1-8B           | 72.7|
> | LxzGordon/URM-LLaMa-3.1-8B                    | 70.1|
> | NCSOFT/Llama-3-OffsetBias-RM-8B               | 69.9|
> | Ray2333/GRM-Llama3-8B-rewardmodel-ft          | 69.5|
> | Ray2333/GRM-llama3-8B-sftreg                  | 68.4|
> | Ray2333/GRM-llama3-8B-distill                 | 67.9|
> | internlm/internlm2-20b-reward                 | 66.3|
> | internlm/internlm2-7b-reward                  | 66.3|
> | LxzGordon/URM-LLaMa-3-8B                      | 65.5|
>
> The Spearman correlation for the rank of these models between the GPT4o-based RM-Bench and Gemini-based RM-Bench is **0.833 with p-value p<0.05**, indicating consistent results.
>
> This consistency reflects that the inherent bias introduced by GPT4o in RM-Bench is not significant. This may be due to the following:
> - First: the evaluation bias concern of using GPT4o mainly occurs when the GPT4o is used for both generating test content and subsequent evaluation [1,2] but in RM-Bench, the GPT4o only serves as the test content generator, the evaluation is done by the objective metrics (e.g., the correctness of the response), which is not affected by the bias of GPT4o.
> - Second: so far (2024-11-20), GPT4o is the most powerful LLM (expecting the reasoning specific model like o1) we have, and it achieves both top 1 in Chatbot Arena [3] under w/o style control setting. This indicates that the style and substance are well balanced in GPT4o, thus the bias introduced by GPT4o is not significant.
>
> We are actively testing more reward models with Gemini-based RM-Bench and will provide updated results in the camera-ready version upon acceptance.
>
> [1] Large Language Models are not Fair Evaluators: https://arxiv.org/abs/2305.17926
>
> [2] Judging LLM-as-a-Judge with MT-Bench and Chatbot Arena: https://arxiv.org/abs/2306.05685
>
> [3] Leaderboard of Chatbot Arena: https://lmarena.ai/
>
>
> ---
>
> > **Weakness 2:** Requiring one more model (maybe from Qwen or LLaMA family) for the correlation analysis.
>
> To address this, we incorporated the **LLaMA-3-8B** model into our analysis. Specifically:
> - LLaMA-3-8B was trained on the Tulu-V2 dataset to obtain an SFT variant. Then, it was trained on the UltraFeedback dataset with the DPO method and Bradley-Terry method to create a DPO variant and a Sequence Classifier Reward Model variant. Finally, with the Sequence Classifier Reward Model variant, we trained a PPO variant of LLaMA-3-8B.
>
> The following table presents the updated results in Table 4:
>
> |                  | LLaMA-3-8B |
> |-|-|
> | DPO (Ref. Model Free) | 58.8   |
> | Seq. Classifier       | 59.8   |
> | DPO (With Ref. Model) | 62.0   |
>
> As shown, the DPO variant of LLaMA-3-8B outperforms the sequence classifier, and performance decreases when the reference model is not used. This observation aligns with the results in the original Table 4.
>
> - For Section 5 results, we also added LLaMA-3-8B results to the correlation figures. These updated figures are included in Appendix M.
> - In summary, after incorporating LLaMA-3-8B results, the positive correlation between RM-BENCH and policy model performance remains intact, further supporting the conclusions in Section 5.
>
> ---
>
> > **Weakness 3:** Missing details about prompts and quality checks in the appendix.
>
> Thank you for this suggestion! To enhance reproducibility, we have included the prompts used in Appendix L.
>
> For quality assurance:
> - **Tasks with Golden Answers (Math & Code)**: We ensure compact responses still contain the correct answers. For Math, this means providing the golden answer; for Code, ensuring the correct solution snippet is included.
> - **Chat Domain Tasks**: Human annotators were ask to
>   1. Verify that error information is successfully injected into responses.
>   2. Confirm that the original correct information is completely replaced by error information.
>   3. Ensure all chosen/rejected responses convey the same meanings.
> - **Safety Domain Tasks**: Since the key in safety is about rejecting or not rejecting queries from users, we use a keyword table that includes terms like "Sorry", "not engage", "can not assistant" and so on to filter out some data points, then we finally manually scan out the dataset for doubt check the correctness of benchmark

---

> ### Author Response · Authors · 2024-11-22
> **Response to Reviewer SzZ1 (Part 2/2)**
>
> > **Weakness 4:** Clarification on length control in Table 2.
>
> We would like to clarify that there is no specific length target for each type of response.
> The difference between a detailed response and a concise response is that the detailed response contains more information, explanation, and details, while the concise response only contains the key information to answer the query.
>
> It is natural that different queries may result in concise responses of varying lengths.
> For example, a concise response to "How to print hello world with Python?" is shorter than a concise response to "Give me a bubble sort code snippet."
>
> This definition aligns with the common practices in communities like Chatbot Arena [1], where they define "concise" means clear but not verbose or excessive.
>
> For quality ensurement, please refer to the response to Weakness 3.
>
> [1] Chatbot Arena: An Open Platform for Evaluating LLMs by Human Preference: https://arxiv.org/abs/2403.04132
>
> ---
>
> > **Question 1:** Overlap with RewardBench.
>
> The user queries in RM-Bench are sourced from RewardBench, but all responses are freshly generated using GPT4o to ensure high-quality, controlled response pairs.
>
> These new-generated responses allow us to conduct substance and style control evaluations, which are not available in RewardBench.
>
> ---
>
> > **Question 2:** Clarifying token numbers of short and long responses in Math and Code domains.
>
> We would like to clarify that the average token of short and long responses in Math and Code are far from identical as Table 2 shows.
> :
> - For Math: Short responses average **319 tokens**, while long responses average **500 tokens** (a 50% increase).
> - For Code: Short responses average **503 tokens**, while long responses average **628 tokens** (a 20% increase).
>
> Besides, considering the token number of response is conditional on the query (different query requires different length response, for example, answer *1+1=?* requires less token number than *solve x^2 -2x + 3 = 0*).
>
> Below, we provide the average ratio of token counts between long and short responses of each query in Math and Code tasks:
>
> |      | Long/Short Ratio |
> |------|-------------------|
> | Math | 1.85              |
> | Code | 1.38              |
>
> This ratio further highlights the substantial difference between long and short responses for each query.
>
> Additionally, as we originally stated in Lines 429–430: "*Nemotron-340B-Reward struggles to disentangle correctness from other factors*," here we mainly want to emphasize the overlap in **correctness** between the chosen and rejected.
> We would further refine this part in the revised version to avoid confusion.
>
> ---
>
> > **Question 3:** Data points in Figure 4.
>
> We would like to clarify that the four data points signify stable increases in reward model performance, resulting in a rank Spearman correlation of **1.0 with p-value p<0.05**.
>
> And Yes, the four points correspond to models trained on different preference datasets as stated in Lines 438–444. From left to right, the datasets are:
> 1. Chatbot Arena 2023
> 2. Nectar
> 3. StackExchange
> 4. HH-RLHF
>
> This information has been added to Figure 4 of Section 5 in the revised version.
>
> ---
>
> > **Question 4:** Request for additional graphs based on Code Hard or Code Normal accuracy.
>
> As we have mentioned above, the preference data is not only StackExchange, but also includes other datasets from general domains like Chatbot Arena 2023, HH-RLHF, and Nectar.
> Considering the corresponding policy model benchmark here is Auto Arena Hard, a general domain benchmark.
> Thus, we would like to clarify that it may not be appropriate to use Code Normal accuracy as the x-axis.
>
> For additional model results, please refer to the response to Weakness 2.
>
> ---
>
> > **Question 5:** Clarification on the y-axis in Figure 8 (Appendix H).
>
> The y-axis represents the relative improvement of the best-of-n results over naive greedy search decoding.
>
> ---
>
> > **Typos:**
>
> We have corrected the typographical errors and appreciate you bringing them to our attention.
> For the omitted sentence on Page 4 Lines 182–183, we have completed the sentence for clarity. Here we mean "Wary LLM is used to generate refusal to benign queries, and Unc LLM is used to generate harmful responses to dangerous queries."

---

> > ### Comment · Reviewer_SzZ1 · 2024-11-24
> > **Thanks for rebuttal**
> >
> > Thank you for addressing my concerns and for adding the necessary details to improve the quality of this submission. I also appreciate the authors' commitment to fully open-sourcing the benchmark along with model weights as confirmed by them in their rebuttal to Reviewer 9kQk. I feel that the community would benefit from the release of such benchmark and hence I have updated my score to Accept.

---

> > > ### Author Response · Authors · 2024-11-25
> > >
> > > Thank you for your active feedback and engagement during the rebuttal process. We appreciate your endorsement!

---

### Author Response · Authors · 2024-11-22
**General Response to Reviewers**

We are grateful to the reviewers for their detailed and constructive feedback on our submission.

We are pleased that all reviewers **unanimously recognized the originality and significance of our work**. To the best of our knowledge, our benchmark, RM-Bench, is **the first** systematic evaluation framework designed to assess reward models' **sensitivity to subtle substance changes** and **robustness to style bias**.

Compared to prior work such as RewardBench, we incorporate both **substance control** and **style control**, and introduce a **Style-Substance Evaluation Matrix** to comprehensively assess reward models.

Additionally, all reviewers **agreed that our framework effectively highlights the shortcomings of existing reward models in these two critical aspects**.

Recent efforts, such as Style-Controlled Chatbot Arena [1] and Length-Controlled AlpacaEval [2], emphasize style control in evaluating policy models to enhance policy models' helpfulness and capability, rather than prioritizing stylistic appeal over substance.

A critical yet overlooked issue is the reward model, which guides policy model training. Failures to detect substantive errors or style bias in the reward model can propagate to policy models, leading to superficial outputs that compromise utility.

Despite its importance, the sensitivity of reward models to substantive errors and their robustness to style bias remain underexplored.
We believe that our work serves as a strong starting point to fill this gap, and we are excited to see the community’s response to our contribution.

[1] *Does Style Matter? Disentangling Style and Substance in Chatbot Arena.* https://lmsys.org/blog/2024-08-28-style-control/

[2] *Length-Controlled AlpacaEval: A Simple Way to Debias Automatic Evaluators.* https://arxiv.org/abs/2404.04475

---

We address a common concern below:

> Prompt for Style Control (Reviewer SzZ1, RBmr)

We thank the reviewers for their suggestions regarding prompts for removing markdown tags and generating concise responses.
We have included these details in **Appendix L** in the revised version of our paper.

---

### Meta-Review · Area_Chair_Mq4J · 2024-12-22

**Metareview:**

This paper introduces a new benchmark for reward modeling that is designed to be robust to subtleties in generated text. In previous benchmarks, rejected answers were usually generated by weak models, which could be easily exploited.

Strengths:
This paper studies an important problem with novel solution ideas. The results provide valuable new observations to the community, particularly regarding DPO models as reward models.

Weaknesses:
The initial version of this paper relied heavily on GPT4o, and many results were conducted on only one model series. However, the authors addressed these concerns by adding additional experimental results during the rebuttal period.

Overall, I think the results of this paper are interesting and significant.

**Additional Comments On Reviewer Discussion:**

This paper provided additional experimental results during the rebuttal period and addressed the concerns from the reviewers very well.

---

### Decision · Program_Chairs · 2025-01-22

Accept (Oral)